# Circulating Tumor Cells and the Non-Touch Isolation Technique in Surgery for Non-Small-Cell Lung Cancer

**DOI:** 10.3390/cancers14061448

**Published:** 2022-03-11

**Authors:** Hiroyuki Adachi, Hiroyuki Ito, Noriyoshi Sawabata

**Affiliations:** 1Department of Thoracic Surgery, Kanagawa Cancer Center, Yokohama 241-8515, Japan; h-ito@kcch.jp; 2Department of Thoracic and Cardiovascular Surgery, School of Medicine, Nara Medical University, Kashihara 634-8521, Japan; nsawabata@naramed-u.ac.jp

**Keywords:** non-touch isolation technique, circulating tumor cell, non-small-cell lung cancer, PV first lobectomy, surgery

## Abstract

**Simple Summary:**

Our review discusses findings for the non-touch isolation technique in surgery, especially in surgery for non-small-cell lung cancer. This technique aims to prevent the release of circulating tumor cells (CTCs) from the tumor nest to the bloodstream during surgery, but its efficacy has not been clearly verified. We have summarized the history of CTC detection, relevance of CTCs to clinical practice, and evidence regarding this technique for lung cancer surgery.

**Abstract:**

Circulating tumor cells (CTCs) are dislodged from the primary tumor into the bloodstream, travel within the bloodstream to distant organs, and finally extravasate and proliferate as epithelial metastatic deposits. The relationship between the existence of CTCs and tumor prognosis has been demonstrated by many researchers. In surgery for malignancies, the surgical manipulation of tumors and tissues around the tumor may lead to the release of CTCs into the bloodstream. The non-touch isolation technique (NTIT) has been advocated to prevent the release of CTCs during surgery. The concept of NTIT is the prevention of intraoperative increment of CTCs from the primary tumor by the early blockade of outflow vessels, and ‘pulmonary vein (PV)-first lobectomy’ during surgery for non-small-cell lung cancer (NSCLC) corresponds to this technique. The concept of PV-first lobectomy is well known among thoracic surgeons, but evidence of its efficacy for preventing the increase of intra- and postoperative CTCs and for improving postoperative prognosis is still uncertain. Our study summarizes evidence regarding the relationship between NTIT and CTCs in NSCLC and suggests the need for further research on CTCs and CTC-detecting modalities.

## 1. Introduction

Almost all patients with solid malignant tumors die from distant metastases. Recently, the outcomes and prognoses after treatment for localized malignant tumors have improved due to better diagnostic modalities that lead to the early detection of malignant tumors. However, the prognosis of patients with metastases remains poor. For example, the 5-year overall survival (OS) of patients with advanced lung cancer is reported to be approximately 1–2 years, even after treatment with molecular-targeted therapies [1]. “Distant metastasis” is thought to occur when tumor cells are dislodged from the primary cancer lesion, spread through the blood stream and/or lymphatic drainage system, affix to other organs, and proliferate. The detection of such tumor cells was first reported in 1869. Ashworth first observed circulating tumor cells (CTCs) in the blood of a man with metastatic cancer using a microscope [2]. Subsequently, some researchers reported the presence of CTCs in the blood stream [3,4,5,6,7,8,9,10], but research on the influence of CTCs on cancer prognosis did not progress even in the 2000s because of their rarity. Because CTCs exist only in low numbers in the blood stream (typically 1–10/mm^3^), they are difficult to detect and capture using conventional technologies such as microscopy. Recent progress in biomedical technology has led to the detection of CTCs in the bloodstream [11], and researchers have reported the prognostic influence of CTCs in colorectal [12], lung [13], breast [14], pancreatic [15], head and neck [16], and prostate [17] cancer.

The mechanism of epithelial cancer metastasis has been explained by the “epithelial-to-mesenchymal transition (EMT)” and “mesenchymal-to-epithelial transition (MET)” theory. In this theory, a series of sequential steps is involved. First, the EMT of individual cells occurs within the primary tumor, leading to their intravasation into the bloodstream, survival and travel as CTCs within the bloodstream, and finally, extravasation at distant sites, where MET culminates in their proliferation as epithelial metastatic deposits [18]. Yu et al. [19] demonstrated EMT in human breast cancer cells in the circulation. However, Yu et al. [19] also reported that CTCs that become resistant to chemotherapy had a very high appearance of markers representing the mesenchymal system and also showed an epithelium appearance. It is difficult to explain this phenomenon using only the EMT/MET theory in single-shot CTCs. To resolve this controversy, the “hybrid EMT/MET” theory in cluster CTCs has become noteworthy. Aceto et al. [20] showed the presence of CTC clusters in the blood stream and their important influence on patient prognosis. They showed that CTC clusters have a 23 to 50-fold increased metastatic potential compared to single CTCs. CTC clusters are composed of 2–50 cancer cells that occur as tumor-derived microemboli that may break off from primary tumors [20]. CTC clusters show various EMT/MET statuses, which consequently demonstrate adherence, migration, anti-anoikis, and tumor formation [21]. According to these results, CTC clusters may be the “distant metastases”. Such progress in research on CTCs as a prognostic factor has attracted the attention of many researchers.

## 2. Non-Touch Isolation Technique in Surgery for Other Malignancies

As mentioned above, CTCs are released from the primary tumor into the bloodstream, especially in CTC clusters, which have a high metastatic potential and are released as tumorous microemboli. Theoretically, this release may be promoted by perioperative procedures, such as preoperative tumor biopsy and intraoperative manipulation. Some researchers have reported an increase in the number of CTCs after surgical manipulation in colorectal [22,23], hepatocellular [24], pancreatic [25], and lung [26,27] cancer. To avoid the release of CTCs by intraoperative surgical manipulation, a non-touch isolation technique (NTIT) has been devised for various cancer surgeries. The concept of NTIT is the prevention of the intraoperative increase in CTCs from primary tumors by the early blockade of outflow vessels.

NTIT was first described for colorectal cancer surgery in 1952. Barnes advocated the ligation of the vascular pedicles and division of the bowel prior to handling the cancer-bearing segment [28]. Thereafter, Turnbull et al. [29] reported the results of a retrospective study evaluating the long-term outcomes of patients who underwent colorectal cancer surgery with NTIT. Their study included 896 patients who underwent colorectal cancer surgery between 1950 and 1964 and showed better outcomes in patients with NTIT than in those without NTIT (5-year OS of 50.86% in patients with NTIT and 34.82% in patients without NTIT). To investigate the efficacy of NTIT, a randomized controlled trial (RCT) was conducted in the 1980s, but this trial could not show any statistical significance regarding NTIT due to the lack of statistical power [30]; therefore, NTIT is not yet a standard procedure for colorectal cancer. However, several studies have shown the efficacy of NTIT in colorectal cancer surgery [31]. An RCT with a large sample size that aims to evaluate the efficacy of NTIT for colorectal cancer surgery has been conducted in Japan [32]. The enrollment for this study has been completed, and the results will be released in the near future.

NTIT has also been adopted for surgery for pancreatic cancers and hepatocellular carcinoma. In the 1990s, Nakao et al. [33] first reported the use of NTIT for pancreatic head cancer using an antithrombogenic portal vein bypass catheter between the mesenteric and intrahepatic portal veins.

Kobayashi et al. proposed the use of NTIT for pancreatoduodenectomy without removing the portal vein for periampullary carcinoma [34]. Hirota et al. [35] proposed NTIT for pancreatoduodenectomy using a hanging-up and clamping technique. Hirota et al. [36] also reported that the 3-year OS rate after surgery for patients with NTIT was superior to that for patients without NTIT (75% and 14%, respectively). However, Gall et al. [37] showed no impact of NTIT on postoperative survival in pancreatic cancer, despite the significant decrease in the number of CTCs detected in the portal vein with NTIT. RCTs evaluating the efficacy of NTIT for pancreatic cancer have been conducted because of the rarity of surgery for pancreatic cancer, meaning that the efficacy of NTIT for pancreatic cancer is yet to be established.

## 3. Non-Touch Isolation Technique in Surgery for Non-Small-Cell Lung Cancer

Regarding surgery for non-small-cell lung cancer (NSCLC), “pulmonary vein (PV)-first lobectomy” is regarded as NTIT. Anatomical lung resection, including lobectomy, which is the standard procedure for the treatment of NSCLC, requires the dissection of the lobar branch of the pulmonary artery (PA), PV, and bronchus. These structures have a complex three-dimensional location; thus, lung manipulation is required for anatomical lung resection. Regarding other types of cancers, thoracic surgeons and researchers, for many years, have thought that surgical manipulation results in the spillage of tumor cells from primary tumors, and many researchers have demonstrated an increase in the number of CTCs or surrogate substances in peripheral blood and the PV after surgical procedures for NSCLC (Table 1). PV-first lobectomy, which is characterized by early PV ligation, has been advocated since the 1950s [38], as it theoretically prevents the outflow of CTCs produced by surgical manipulation to the circulation. This is because the PV is the discharge canal for the central bloodstream; thus, PV-first lobectomy improves postoperative prognosis. To evaluate the efficacy of PV-first lobectomy, researchers recently attempted to demonstrate its effect on the behavior of perioperative CTCs and postoperative prognoses. For the analysis of perioperative changes in CTC after PV-first lobectomy, Kurusu et al. [39] conducted an RCT in a small cohort dividing patients into two groups (PA-first ligation and PV-first ligation) and showed that patients in the PV-first group had a lower positive conversion rate for peripheral arterial carcinoembryonic antigen (CEA) mRNA expression than patients in the PA-first group. Song et al. [40] showed in their RCT that the mRNA expression of CD44v6 and CK19 in the PA-first group increased after vessel ligation compared to before ligation, whereas the expressions in the PV-first group were similar before and after vessel ligation. Duan et al. [41] and Wei et al. [42] also reported the efficacy of the PV-first procedure in preventing CTC production. However, Ge et al. [43] conducted an RCT focused on the detection of CK19 and CEA mRNA in peripheral blood and reported no significant difference between the PV-first and PA-first groups in the trend of change of the perioperative values of CEA mRNA. Moreover, they detected circulating epithelial cells (supposed non-malignant bronchial epithelial cells entering into the bloodstream by surgical manipulation) in two patients in the control group who underwent surgery for non-malignant lesions, which suggests that CK19 mRNA might be detected even in the peripheral blood of patients without malignancies. Hashimoto et al. [44] also reported that the PV-first procedure had a significant influence on the intraoperative increase in CTCs, and the role of PV-first lobectomy in preventing the increment of CTCs is still uncertain (Table 2).

To analyze the effect of PV-first lobectomy on postoperative prognosis, we systematically reviewed the literature. The selection of studies was based on the titles, abstracts, and full papers, with the following inclusion criteria: (1) comparative studies examining PV-first versus PA-first lobectomy, (2) RCTs or observational retrospective/prospective cohort and case-control studies, and (3) studies that reported postoperative prognostic outcomes, such as recurrence and survival rates. The literature search was conducted using PubMed with search terms: (“lung cancer” OR “lung carcinoma” OR “lung neoplasm”) AND (“vein-first” OR “artery-first” OR “vessel ligation” OR “vessel sequence”) AND (“surgery” OR “operat*” OR “postoperative”), and 46 studies were identified. Among them, six studies were included (Figure 1; Table 3). Refaely et al. [55] reported in their retrospective study that the sequence of vessel interruption (PV-first or PA-first) had no influence on disease recurrence. Kozak et al. [56] also reported the non-efficacy of the sequence of vessel interruption regarding postoperative survival in their RCT. However, their studies included patients who underwent lobectomy through open thoracotomy, which requires more aggressive manual manipulation than VATS lobectomy and may lead to the release of large amounts of CTCs in the early phase of surgery before PV ligation. Li et al. [57] reported the non-efficacy of PV-first lobectomy for postoperative survival in their study, which included only patients who underwent lobectomy by VATS, but their analyses were conducted with patient classification into three groups (PV-first, PA-first, and PA-PV-PA sequence), and no data comparing PV-first and PA-first alone were shown. However, recent studies that included only VATS lobectomy showed the efficacy of PV-first lobectomy for postoperative survival. Sumitomo et al. [58] reported in their retrospective cohort study that PV-first was an independent prognostic factor for better disease-free survival, and He et al. [59] reported in their retrospective study that PV-first lobectomy through the VATS approach was preferred for patients with squamous cell carcinoma, which seemed to metastasize through the bloodstream rather than the lymphatic stream. Moreover, Wei et al. [47] conducted a retrospective cohort study with propensity score matching to minimize selection bias and reported that the 5-year OS rate in the PV-first group was significantly better than that in the PA-first group (73.5% in PV-first vs. 57.6% in PA-first, *p* = 0.002). In addition, the 5-year disease-free survival and 5-year lung cancer-specific survival in the PV-first group were significantly better than those in the PA-first group. Huang et al. [60] conducted a meta-analysis of five studies, including those by Kozak et al. [56], Li et al. [57], Sumitomo et al. [58], He et al. [59], and Wei et al. [47], and concluded that PV-first ligation is recommended during lobectomy for patients with NSCLC whenever possible. Based on these results, we consider that PV-first lobectomy through the VATS approach has the potential to improve postoperative survival in patients with surgically resectable NSCLC. However, only a few studies have evaluated the relationship between the PV-first technique and postoperative survival. Large-scale, prospective, and multicenter RCTs are needed to clarify the efficacy of PV-first lobectomy.

## 4. Conclusions

Surgical manipulation has the potential to induce increments in intra- and postoperative CTCs. NTIT, which is considered a PV-first technique in surgery for NSCLC, theoretically prevents the release of these CTCs by early blockade of outflow vessels. However, evidence of the efficacy of NTIT in the prevention of CTC release and postoperative prognosis is still unsatisfactory. This may be partially due to insufficient knowledge regarding the biological behavior of CTCs and the lack of technologies to detect CTCs. We believe that increased research into the biology of CTCs and the technologies for detecting CTCs will accelerate the interest of clinical physicians, which will lead to other clinical studies and contribute to improved patient prognosis.

## Figures and Tables

**Figure 1 cancers-14-01448-f001:**
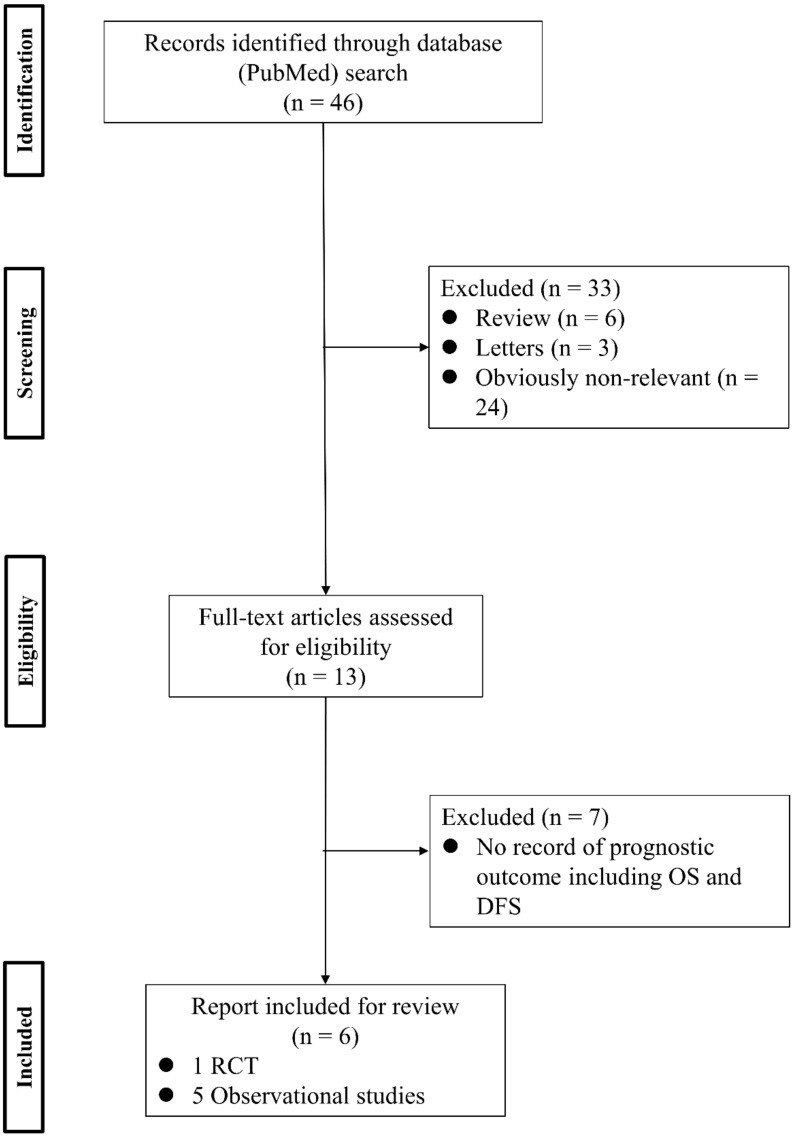
Flowchart of the retrieval of relevant studies. DFS: disease-free survival, OS: overall survival, RCT: randomized controlled trial.

**Table 1 cancers-14-01448-t001:** Summary of studies researching the relationship between surgical manipulation and change in CTCs detection in non-small-cell lung cancer.

Authors	Year	*N*	Detected Factors	Device or Method for CTCs Detection	Results
Yamashita et al. [45]	2000	29	CEA mRNA in peripheral blood before/after lobectomy	RT-PCR	Of the 29 patients, the preoperative blood samples from 18 patients were negative for CEA mRNA. Of these 18 patients, 16 (89%) were positive for CEA mRNA during surgery, although the remaining 2 patients (11%) were consistently negative for CEA mRNA.
Sawabata et al. [46]	2007	9	CTCs in peripheral blood before/after lobectomy	CellSearch system	3 patients (including 1 patient with preoperative detection of CTCs) showed CTCs after surgery. No CTCs were detected in any patient 10 days after surgery.
Hashimoto et al. [44]	2014	30	CTCs in peripheral artery and PV before/after lobectomy	CellSearch system	The CTC count in PV was significantly increased (median 60.0 cells/2.5 mL) after surgical manipulation. The increase of CTCs in PV was not associated with the sequence of vessel ligation.
Sawabata et al. [47]	2016	23	CTCs (single or cluster) in peripheral artery and resected PV	ScreenCell^®^ CTC selection kit	CTCs were detected from both the artery and PV (8 as single and 2 as cluster) in 10 of 16 patients (62.5%) who showed no CTCs before surgery. All 7 patients with CTCs (1 as single and 6 as cluster) before surgery showed cluster CTCs in both artery and PV. Detection of cluster CTCs after surgery was a significant factor for a worse prognosis.
Huang et al. [48]	2016	79	CTCs in peripheral blood before/after lobectomy	Antibodies for epithelial marker expression	30 of the 79 patients tested positive for CTCs before surgery (37.97%). The increase in number of CTCs from before surgery to after surgery was significantly lower in the VATS group than in the conventional thoracotomy group.
Reddy et al. [49]	2016	32	CTCs in peripheral vein and PV before/after lung resection	Antibodies for epithelial marker expression	20 patients had 1 or more CTCs in at least 1 sample. The mean number of CTCs from the peripheral vein at the preoperative, intraoperative, and postoperative periods was 1.3, 1.9, and 0.6, respectively. The number of CTCs in PV was significantly higher when preoperative bronchoscopic biopsy was performed.
Matsutani et al. [27]	2017	31	CTCs in peripheral artery before/after lobectomy	ScreenCell^®^ CTC selection kit	There were 13 pre-CTC(+) patients and 17 post-CTC(+) patients. Among the 16 patients who were pre-CTC(-), 4 were eventually post-CTC(+), while all pre-CTC(+) patients remained post-CTC(+).
Murlidhar et al. [50]	2017	36	CTCs in peripheral vein and PV before/after lung resection	OncoBean Chip	Preoperatively and intraoperatively, PV had a significantly higher number of CTCs compared with the peripheral vein. Long-term surveillance indicated that presence of cluster CTCs in preoperative peripheral blood predicted a trend toward a poor prognosis.
Hu et al. [51]	2017	168	cfDNA in peripheral blood after lung resection	Tiangen Serum/Plasma Circulating DNA Kit	5 patients with recurrence in 4 months had significantly higher circulating cfDNA at 30 days after surgery. 6 patients with recurrence after 4 months and 5 patients without recurrence demonstrated significantly lower circulating cfDNA.
Duan et al. [41]	2019	33	CTCs in PV before/after lobectomy	oHSV1-hTERT-GFP method	The CTC detection rate before PV interruption was 79.0%, while the rate after lobectomy was 100%. The CTC count was also significantly higher following surgery.
Wei et al. [42]	2019	78	CTCs in RA before/after lobectomy	FR^+^CTCs Detection Kit	Incremental change in CTCs between before and after lobectomy was observed in 65.0% of cases in the artery-first group and 31.6% of cases in the vein-first group.
Tamminga et al. [52]	2020	31	CTCs in RA and PA/PV before/after lung resection	CellSearch system	CTCs were more often detected in the PV (70%) than in the RA (22%). After surgery, the RA, but not the PV, showed significantly less often CTCs.
Sawabata et al. [53]	2020	81	CTCs in peripheral blood before/after lung resection	ScreenCell^®^ CTC selection kit	Among 81 lung cancer patients with negative preoperative results for CTCs, no CTC was found in 58 (71.6%), only a single CTC was found in 6 patients (7.4%), and CTC clusters were found in 17 patients (21.0%) postoperatively. Multivariate analysis revealed that recurrence was independently related to the postoperative detection of single CTCs and CTC clusters.
Katopodis et al. [54]	2021	54	CTCs and cfDNA in peripheral vein before/after lung resection	ImageStream™ (for CTCs) Maxwell RSC ccfDNA Plasma Kit (for cfDNA)	CTCs were increased in postoperative blood samples in the 54 patients. Patients who underwent thoracotomy instead of VATS had higher numbers of CTCs postoperatively. cfDNA were also significantly increased in postoperative samples.

CEA, carcinoembryonic antigen, cfDNA: cell-free DNA, CTCs: circulating tumor cells, PA: pulmonary artery, PV: pulmonary vein, RA: radial artery, RT-PCR: reverse transcription polymerase chain reaction, VATS: video-assisted thoracic surgery.

**Table 2 cancers-14-01448-t002:** Summary of studies investigating the relationship between the PV-first technique and change in postoperative CTCs detection in non-small-cell lung cancer.

Authors	Years	Study Design	*N*	Detected Factors	Device or Method for CTCs Detection	Results	Conclusion ^†^
Kurusu et al. [39]	1998	Prospective randomized study	36	CEA mRNA in peripheral artery	RT-PCR	16/30 patients with NSCLC (53.3%) and 5/6 patients with SCLC (83.3%) showed positivity for CEA mRNA before surgery. Among 14 patients with NSCLC who were negative before surgery, patients in the PA-first group showed a higher rate of positive conversion than patients in the PV-first group (6/7 patients (85.7%) in the PA-first group vs. 3/7 patients (42.9%) in the PV-first group).	Effective
Ge et al. [43]	2006	Prospective randomized study	23	CK19 and CEA mRNA in peripheral blood	RT-PCR	The values of CK19 mRNA in blood during surgery in the PA-first group was non-significantly higher than that in the PV-first group. The values of CEA mRNA in blood gradually increased from the preoperative to postoperative period in both the PV-first and PA-first groups, but the increase ratio was slightly larger in the PA-first group than in the PV-first group (difference not significant).	Uncertain
Song et al. [40]	2013	Prospective randomized study	30	CK19 and CD44v6 mRNA in proximal PV	real-time PCR	In the PA-first group, the mRNA expressions of CD44v6 and CK19 were significantly higher after ligation than before ligation. In the PV-first group, the mRNA expressions of CD44v6 and CK19 were similar before and after ligation.	Effective
Hashimoto et al. [44]	2014	Prospective cohort study	30	CTCs in peripheral artery and PV	CellSearch system	The CTC count in PV was significantly increased (median 60.0 cells/2.5 mL) after surgical manipulation. The increase in CTCs in PV after lobectomy was not associated with the sequence of vessel ligation.	Not effective
Duan et al. [41]	2019	Prospective cohort study	33	CD45-GFP^+^CTCs in PV before/after lobectomy	oHSV1-hTERT-GFP method	The post-CTC count was significantly higher in patients in whom the PV was interrupted prior to the PA (15 counts) than in patients in whom the PA was interrupted before the PV (7 counts).	Effective
Wei et al. [42]	2019	Prospective randomized study	86	FR^+^CTCs in peripheral artery	FR^+^CTCs Detection Kit	8 patients were not included because postoperative blood samples were not collected. Incremental change in FR^+^CTCs was observed in 26/40 patients (65.0%) in the PA-first group and in 12/38 patients (31.6%) in the PV-first group. Multivariate analysis revealed that the PA-first procedure was an independent risk factor for increase in FR^+^CTCs during surgery (HR 4.03 (95%CI, 1.53–10.63))	Effective

^†^ Authors’ conclusion whether a PV-first approach was effective for preventing the release of CTCs into the blood stream; CEA, carcinoembryonic antigen, cfDNA: cell-free DNA, CI: confidence interval, CTCs: circulating tumor cells, HR: hazard ratio, PA: pulmonary artery, PV: pulmonary vein, RA: radial artery, RT-PCR: reverse transcription polymerase chain reaction, VATS: video-assisted thoracic surgery.

**Table 3 cancers-14-01448-t003:** Summary of studies researching relation between PV-first technique and postoperative prognosis in non-small-cell lung cancer.

Authors	Years	Design	Evidence Level *	n	Approach	Stage	F/U Period (Month)	Prognostic Outcome			Additionals	Conclusion ^†^
								5y-OS (V-first vs. A-first)	5y-DFS (V-first vs. A-first)	Others		
Refaely et al. [55]	2003	Ret	4	279 (V-first 133, A-first 146)	Open or VATS	I–IV	22.6 (mean)	NA	NA	Rec rate; 51% in V-first 53% in A-first (*p* = 0.70)	NS in multivariate analysis on disease recurrence (OR 1.29 [95%CI 0.73-2.29])	Not effective
Kozak et al. [56]	2013	Single-center RCT	2	385 (V-first 170, A-first 215)	Open or VATS	I–III	63 (median)	54% vs. 50% (*p* = 0.82)	NA	-	NS in cancer related deaths (*p* = 0.67) and in non-cancer related deaths (*p* = 0.26).	Not effective
Li et al. [57]	2015	Ret	4	334 (V-first 174, A-first 93, A-V-A 67)	VATS	I–II	30 (V-first) 26 (A-first) 20 (others) (median)	NS in 1-, 3-, 5-OS (*p* > 0.05)	NS in 1-, 3-, 5-DFS (*p* > 0.05)	Ns in local Rec and distant Metas (*p* > 0.05)	-	Not effective
Sumitomo et al. [58]	2018	Ret	4	187 (V-first 104, A-first 83)	VATS	I–IIIA (exc. AIS & MIA)	54.9 (median)	90.9% vs. 82.7% (*p* = 0.080)	88.2% vs. 75.7% (*p* = 0.019)	-	DFS was significantly longer in V-first among stage I, but not significant among stage II–IIIA. A-first was independent poor prognostic factor for DFS in multivariate analysis. (HR 2.127 [95%CI 1.009-4.481])	Effective
He et al. [59]	2019	Ret	4	60 (V-first 33, A-first 27)	VATS	I–IVA	NA	66.67% vs. 44.44% (*p* = 0.056)	39.40% vs. 29.63% (*p* = 0.176)	CS deaths; 14/33 in V-first 17/27 in A-first (*p* = 0.227)	Statistically significant difference was shown in OS and DFS among Sq patients. (70.0% vs. 25.0% in OS, and 40.0% vs. 0% in DFS)	Partially Effective
Wei et al. [47]	2019	Ret (PS matching)	4	420 (V-first 210, A-first 210)	VATS	I–II (tumor > = 2 cm)	30 (median)	73.6% vs. 57.6% (*p* = 0.002)	64.6% vs. 48.4% (*p* = 0.001)	5y-CS survival; 76.4% in V-first 59.9% in A-first (*p* = 0.002)	A-first was independent poor prognostic factor for OS in multivariate analysis. (HR 1.65 [95%CI 1.07–2.56])	Effective

* According to the Oxford Centre for Evidence-Based Medicine 2011 Levels of Evidence. ^†^ Authors’ conclusion whether PV-first lobectomy was effective or not to improve postoperative prognosis. A: pulmonary artery, AIS: adenocarcinoma in situ, CI: confidential interval, CS: cancer-specific, DFS: disease-free survival, exc.: exclude, F/U: follow up, HR: hazard ratio, MIA: minimally invasive adenocarcinoma, Metas: metastasis, NA: not available, NS: not significant, NSCLC: non-small-cell lung cancer, OR: Odds ratio, OS: overall survival, PS: propensity score, RCT: randomized control study, Rec: recurrence, Ret: retrospective cohort study, Sq: squamous cell carcinoma, V: pulmonary vein, VATS: video-assisted thoracic surgery.

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
