# Peer review of "Circulating Tumor Cells and the Non-Touch Isolation Technique in Surgery for Non-Small-Cell Lung Cancer"

_cancers, 2022, doi:10.3390/cancers14061448_

Round 1
Reviewer 1 Report
This is a review article examining the published studies on pulmonary vein first isolation in surgery for NSCLC to limit the spread of circulating tumor cells (CTC). This topic has merited interest over the years as the theoretical principles seem valid. The authors are clearly interested in this field of research as three of the listed studies in Table 1 were from their group. However, as this article points out, the published data are mixed and definitive conclusions are hard to come by. Overall, this article does not add to our current knowledge but does a good job in summarizing the existing data and offering salient suggestions as to where the future may lead.
Author Response
We wish to express our appreciation to the Reviewer 1 for their insightful comments on our paper. We believe that the comments have helped us significantly improve our manuscript.
To enhance the comprehensiveness of our manuscript, we significantly revised Table 3 (pages 11-12) by adding new columns for levels of evidence, study design, tumor staging, and so on, based on the advice of the reviewers.
In addition, we defined the abbreviations used in Table 3 in a footnote under the table (page 12, lines 205-210). We also revised the titles of Tables 1 and 2 (page 5, line 151 and page 7, line 154, respectively).

Reviewer 2 Report
- In Table 1 & 3, Journal name looks unnecessary as a table content.
- The tables should be re-organized to enhance comprehensiveness. Tables are too descriptive to show the difference between methods. I recommend to merge table 1 and 3 and to insert more columns to show a level of evidence, study design, single center or multi-center, tumor staging, 5 year DFS & OS, HR etc.
- This is submitted as a review article but rather close to meta analysis. I recommend to revise this manuscript as a meta-analysis and add an appropriate results and material and methods.
Author Response
We wish to express our strong appreciation to the Reviewer 2 for his or her insightful comments on our paper. We believe that the comments have helped us significantly improve our manuscript. In particular, we wish to acknowledge the Reviewer’s highly valuable comments on the Table 3.
Comments and Suggestions 1:
In Table 1 & 3, Journal name looks unnecessary as a table content.
Response:
We thank the reviewer for this pertinent comment. Accordingly, we deleted the column containing journal names in Tables 1, 2, and 3.
Comments and Suggestions 2:
The tables should be re-organized to enhance comprehensiveness. Tables are too descriptive to show the difference between methods. I recommend to merge table 1 and 3 and to insert more columns to show a level of evidence, study design, single center or multi-center, tumor staging, 5 year DFS & OS, HR etc.
Response:
We thank the reviewer for their valuable suggestions, and apologize for the descriptiveness of the tables. Table 1 is a summary of the studies that investigated the relationship between surgical manipulation and change in CTC detection, while Table 3 is a summary of the studies that investigated the relation between the PV-first technique and postoperative prognosis. As mentioned in manuscript, evidence for the association between the presence of CTCs and postoperative prognosis remains inconclusive. Thus, we believe that Tables 1 and 3 should remain separate, as in the present manuscript. Instead, we replaced the original title for Table 1, which was “Summary of studies researching the relationship between surgery and CTCs in non-small-cell lung cancer,” page 5, line 151)“Summary of studies that investigated the relationship between surgical manipulation and change in CTCs detection in non-small-cell lung cancer.” For enhanced clarity, we replaced the title for Table 2, which was “Summary of studies investigating the relationship between the PV-first technique and postoperative CTCs in non-small-cell lung cancer,” with the following title (page 7, line 154): “Summary of studies that investigated the relationship between the PV-first technique and change in postoperative CTCs detection in non-small-cell lung cancer.”
As per the reviewer’s suggestion, to enhance the comprehensiveness of our manuscript, we significantly revised the Table 3 (pages 11-12), and added new columns for level of evidence, study design, tumor staging, and so on. In addition, we defined the abbreviations used in Table 3 in a footnote under the table (page 12, lines 205-210).
Comments and Suggestions 3:
This is submitted as a review article but rather close to meta analysis. I recommend to revise this manuscript as a meta-analysis and add an appropriate results and material and methods.
Response:
We wish to express our deep appreciation to the reviewer for this insightful comment. As a matter of fact, we initially planned on conducting a meta-analysis regarding the association between the PV-first technique and postoperative prognosis. However, Huang et al. have already conducted a meta-analysis based on the same studies that we retrieved; this was published in 2021 (reference #61 in our manuscript). We believe that a meta-analysis of the change in CTC detection may have been slightly inappropriate for our research topic, because we focused on the relationship between the surgical technique and the postoperative prognosis. Therefore, we submitted this manuscript as a review article to emphasize the importance of further basic and clinical research on CTCs, which will be beneficial for clinical practice. Thus, we wish to retain the style of the article as is.

Reviewer 3 Report
It is an interesting paper reviewing some articles focusing on the manipulation of cancer during surgery and its relation to circulating cancer cell.
The topic is very interesting. The conclusion "there is no evidence that manipulation can reduce circulating cancer cell" is disappointing, but this is status of the literature at the moment. The same for "increased research is needed..."
Although the paper recall attention on an topic a bit neglected in recent years
Author Response
We wish to express our strong appreciation to Reviewer 3 for their insightful comments on our paper. We believe that the comments have helped us significantly improve our manuscript.
To enhance the comprehensiveness of our manuscript, we significantly revised Table 3 (pages 11-12) by adding new columns for levels of evidence, study design, tumor staging, and so on, based on the advice of the reviewers.
In addition, we defined the abbreviations used in Table 3 in a footnote under the table (page 12, lines 205-208). We also revised the titles of Tables 1 and 2 (page 5, line 151 and page 7, line 154, respectively).

Round 2
Reviewer 2 Report
Some revisions for tables are made as requested. However authors should provide methods for data collection, literature search, selection criteria, and statistical analysis for fulfilling minimum format as meta-analysis.
Author Response
 We wish to express our deep appreciation to the reviewer for this insightful comment. As commented on previous revision, we initially planned on conducting a meta-analysis regarding the association between the PV-first technique and postoperative prognosis. However, Huang et al. have already conducted a meta-analysis based on the same studies that we retrieved; this was published in 2021 (reference #61 in our manuscript), and we abandoned our meta-analysis and made our manuscript as “Review article”.
 We submitted this manuscript as a review article in whole to emphasize the importance of further basic and clinical research on CTCs, which will be beneficial for clinical practice. Thus, we wish to retain the style of the article as is.
 A co-author Prof. Sawabata, who is the Editor of the special issue, agree to publish the review article in this style.

Round 3
Reviewer 2 Report
I strongly recommend revise this article as meta-anlysis format.
Author Response
We wish to express our deep appreciation to the reviewer for this insightful comment.
As commented on previous revision, we initially planned on conducting a meta-analysis regarding the association between the PV-first technique and postoperative prognosis. However, Huang et al. have already conducted a meta-analysis based on the same studies that we retrieved; this was published in 2021 (reference #61 in our manuscript), and we abandoned our meta-analysis and made our manuscript as “Review article”.
 We submitted this manuscript as a review article in whole to emphasize the importance of further basic and clinical research on CTCs, which will be beneficial for clinical practice. Thus, we wish to retain the style of the article as is.
 Again, a co-author Prof. Sawabata, who is the Editor of the special issue, agree to publish the review article in this style.
